# Inhibition of *E. coli* RecQ Helicase Activity by Structurally Distinct DNA Lesions: Structure—Function Relationships

**DOI:** 10.3390/ijms232415654

**Published:** 2022-12-09

**Authors:** Ana H. Sales, Vincent Zheng, Maya A. Kenawy, Mark Kakembo, Lu Zhang, Vladimir Shafirovich, Suse Broyde, Nicholas E. Geacintov

**Affiliations:** 1Chemistry Department, New York University, 31 Washington Place, New York, NY 10003-5180, USA; 2Division of Arts and Sciences, NYU Shanghai, 1555 Century Avenue, Shanghai 200122, China; 3Department of Biology, New York University, 31 Washington Place, New York, NY 10003-5180, USA

**Keywords:** RecQ helicase, unwinding, processivities, CPD and (6-4) thymine dimers, benzo[*a*]pyene diol epoxide-DNA adducts, spiroiminodihydantoin

## Abstract

DNA helicase unwinding activity can be inhibited by small molecules and by covalently bound DNA lesions. Little is known about the relationships between the structural features of DNA lesions and their impact on unwinding rates and processivities. Employing *E.coli* RecQ helicase as a model system, and various conformationally defined DNA lesions, the unwinding rate constants *k_obs_* = *k_U_ + k_D_*, and processivities *P* = (k*_U_/(k_U_ + k_D_*) were determined (*k_U_*_,_ unwinding rate constant; *k_D_*, helicase-DNA dissociation rate constant). The highest *k_obs_* values were observed in the case of intercalated benzo[*a*]pyrene (BP)-derived adenine adducts, while *k_obs_* values of guanine adducts with minor groove or base-displaced intercalated adduct conformations were ~10–20 times smaller. Full unwinding was observed in each case with the processivity *P* = 1.0 (100% unwinding). The *k_obs_* values of the non-bulky lesions T(6−4)T, CPD cyclobutane thymine dimers, and a guanine oxidation product, spiroiminodihydantoin (Sp), are up to 20 times greater than some of the bulky adduct values; their unwinding efficiencies are strongly inhibited with processivities *P* = 0.11 (CPD), 0.062 (T(6−4)T), and 0.63 (Sp). These latter observations can be accounted for by correlated decreases in unwinding rate constants and enhancements in the helicase DNA complex dissociation rate constants.

## 1. Introduction

Helicases are motor proteins that convert the free energy of ATP hydrolysis into the mechanical energy required for the separation of double-stranded DNA into two single strands [1]. Helicases are classified into superfamilies SF1 and SF2 [2]. The RecQ helicases, that belong to the SF2 superfamily, are a highly conserved group of DNA helicases with diverse roles in multiple DNA metabolic processes that include DNA recombination, replication, and DNA repair [3]. Defects in RecQ helicases are associated with susceptibilities to cancer and premature aging, genome instability, and hypersensitivity to DNA damaging agents in cultured cells [4,5,6]. The prototypical member of the RecQ protein group is the *E. coli* RecQ helicase, which unwinds double-stranded DNA with a 3′→5′ polarity by an ATP-driven inchworm mechanism [7]. Helicases have long been considered as potential targets in chemotherapeutic applications [8]. However, the relationships between the structural features of helicase inhibitors and their efficiencies as inhibitors of DNA unwinding activity are not well understood.

The objective of this work was to gain new insights into the relationships between the chemical structures of different DNA lesions, their conformations in double-stranded DNA, and their impact on helicase unwinding rates and efficiencies. The helicase *E. coli* RecQ was selected because its structural features and mechanistic properties have been extensively studied and characterized [3,6,7,9,10]. In addition, RecQ is of interest because, unlike many other helicases, it is known to unwind some non-canonical forms of DNA that play important roles in DNA repair, recombination, and replication [11]. Finally, RecQ is known to bind as a monomer to single- and double-stranded DNA substrates with similar affinities [12], thus simplifying the analysis of DNA unwinding kinetics.

The DNA lesions studied include non-bulky DNA lesions generated in human tissues exposed to the UV components of sunlight, and bulky DNA adducts derived from the ubiquitous environmental pollutant benzo[*a*]pyrene (BP), also found in human tissues [13,14]. The bulky DNA adducts selected for this study include those derived from the metabolic activation of BP to reactive diol epoxide derivatives [15]. The covalent binding of the two enantiomeric BP-derived (+)- and (−)-7,8-dihydrodiol-9,10-epoxides (BPDE) forms covalent bonds with the exocyclic amino groups of guanine or adenine in DNA. This covalent bond formation occurs by *cis*- or *trans*-addition of the exocyclic amino groups of guanine or adenine in DNA to the C10 position of BPDE [14]. These reactions generate various stereoisomeric BPDE-*N*^2^-guanine (or *N*^6^-adenine) DNA adducts with different DNA conformations [16]. The bulky BP polycyclic aromatic ring systems assume either intercalative or external minor groove conformations [16,17,18,19,20,21]. The non-bulky DNA lesions studied in this work include the ultraviolet light-induced *cis-syn*-cyclobutane thymine dimer CPD, and the T(6−4)T thymine dimer photolesions, as well as the oxidatively generated spiroiminodihydantoin (Sp) lesion [22]. The chemical structures of these DNA lesions are depicted in Figure 1. It is shown that the strongest inhibition of unwinding activity is caused by the non-bulky, cross-inked thymine dimer lesions. By contrast, bulky polycyclic aromatic DNA adducts are less inhibiting in a manner that depends on their conformations in double-stranded DNA.

## 2. Materials and Methods

### 2.1. Materials

Recombinant *E. coli* RecQ helicase was purchased from the company Abcam. The 2′-oligodeoxynucleotides were purchased from Integrated DNA Technologies (Coralville, IA, USA). The racemic *anti*-BPDE ((+/−)-7*R*,8*S*)-dihydroxy-(9*S*,10*R*)-epoxy-7,8,9,10-tetrahydrobenzo[*a*]pyrene, known as (+)-or (−)-*anti*-BPDE) was originally obtained from the National Cancer Institute Chemical Carcinogen Repository.

### 2.2. DNA Substrates

The bulky DNA adducts (Figure 1) were derived from the reactions of racemic mixtures of *anti-*BPDE with the oligonucleotide 5′-d(CCATCXCTACC) with X = dG or dA. The diol epoxides BPDE react with purines by *cis*- *or trans*-addition of the exocyclic amino groups of guanine or adenine to the C10 carbon atom of BPDE (Figure 1). The modified oligonucleotides were separated from one another and purified as described earlier [20]. The conformations of these DNA adducts embedded in double-stranded oligonucleotides were previously determined by NMR methods [16,23,24].

The thymine dimer (CPD) and the pyrimidine(6−4)pyrimidone T(6−4)T lesions were generated by UV irradiation of the oligonucleotide 5′-d(GCAAGTTGGAG) in aqueous solutions. The oligonucleotide sequences containing the different photoproducts were separated from one another by HPLC methods and further purified by gel electrophoresis, as described earlier [25].

The 32-mer helicase-translocating sequences (Figure 2) containing the lesions were prepared by ligating the 11-mer oligonucleotides with the 5′- and 3′-flanking 8- and 13-mer sequences, 5′-Cy3-GCAGGCAT containing the Cy3 fluorophore, and 5′-GGATCCTCTTTTT. The 32-mer modified sequences obtained by this approach were purified by denaturing PAGE and annealed with 47-mer sequences containing the 22-mer complementary fragments 5′-T_26_CCGGTAGCGATGGATGCCTGC-BHQ2 (Black Hole Quencher) by heating the solution at 90 °C for 5 min, followed by slow cooling to room temperature overnight to form the forked DNA substrates. We designed this forked DNA duplex with a short 10 nucleotide-long 3′-overhang that can accommodate a single RecQ molecule^13^.

### 2.3. Monitoring Unwinding Kinetics by Fluorescence Methods

The RecQ helicase-catalyzed DNA unwinding kinetics were monitored in real time by a fluorescence method [1]. The forked DNA substrates were labelled with the fluorophore (Cy3) at the blunt end, and BHQ2 opposite Cy3 on the opposite strand (Figure 2). The unwinding reactions were initiated by flow-mixing the reagents in buffer solution (20 mM TRIS-HCl, pH 7.6, 10 mM KCl, 5 mM MgCl_2_, 2 mM DTT, 5% glycerol, and 0.1 μg/μL bovine serum albumin) using two manually driven syringes connected through a T-mixer to a quartz cell (3 × 3 mm). In typical experiments, equal volumes (50 μL) of a solution containing the forked DNA substrate and ATP (2 mM) in the first syringe were combined with a solution in the second syringe that contained the RecQ helicase. The time-dependent increase in the fluorescence intensity was monitored by a Photon Technology International Spectrofluorometer (Division of Horiba, Edison, NJ, USA).

The fluorescence of the Cy3 dye was selectively excited with a green diode laser (515 nm, ∼200 μW) and the fluorescence emission was registered at 564 nm by a PC-interfaced photomultiplier (resolution 200 ms/point). The monitoring of the fluorescence signal was initiated within ~1 s after the flow-mixing of the reagents.

Unwinding of the double-stranded DNA substrates results in the formation of free single-stranded oligonucleotides. At the 5 nM DNA concentrations used in most of this work, the re-annealing of Cy3- and BHQ2 single-stranded oligonucleotides was negligible on the time scale of our experiments (Appendix A).

The fluorescence intensity corresponding to fully unwound DNA was measured independently in each unwinding experiment by subjecting an aliquot of the same DNA solution to formamide that resulted in the full unwinding of the double-stranded DNA.

## 3. Results and Discussion

### 3.1. Overview

The effects of non-covalently bound small inhibitors on helicase activities have been extensively studied [8,11]. However, less attention has been focused on the effects of covalently bound DNA adducts [26]. A set of three stereoisomeric bulky BP-G lesions derived from the binding of (+)- or (−)-*anti*-BPDE to the exocyclic amino groups of guanine in DNA, each characterized by remarkable differences in adduct conformations, are shown in Figure 3.

In this work, we used kinetic methods to determine the shapes of the unwinding curves and the unwinding rate constants *k_obs_* using unmodified DNA as a benchmark. We provide novel insights into (1) the magnitudes of helicase unwinding rates and processivities using sets of bulky and non-bulky DNA lesions as substrates, and (2) the relationships between the conformational and structural features of DNA lesions, and their impact on the helicase unwinding rate constants and processivities. The objective of this study was to determine the impact of a variety of conformationally distinct DNA lesions on the unwinding of double-stranded DNA catalyzed by the helicase RecQ [7]. In the following section we summarize the structural and conformational features of the DNA lesions employed in this study.

### 3.2. Determination of Helicase-Catalyzed Unmodified DNA Unwinding Parameters

#### 3.2.1. Kinetic Parameters

Analysis of the unwinding kinetics suggests that the apparent unwinding rate constant *k_obs_* is affected by two factors, the association rate constant for complex formation between RecQ and the DNA substrate (rate constant *k_a_*), and the unwinding processivity. The latter is defined as the probability that a helicase will successfully complete an unwinding step instead of dissociating from the DNA substrate [1,9].

#### 3.2.2. RecQ-Catalyzed Unwinding of Unmodified DNA

Following the injection of ATP into a pre-mixed helicase–unmodified DNA solution with concentrations in the nanomolar range, a rapid burst phase is observable that is attributed to pre-existing helicase DNA complexes as described earlier [9]; this burst phase is followed by a slower unwinding component (Figure 4). The time dependence of the latter component can be followed by monitoring the formation of single-stranded DNA molecules ([DNA]ss) as a function of time and determining *k_obs_* by fitting the experimental data points to the standard exponential equation [9]:ssDNA(t)/ssDNA_tot_ = (1 − exp[−*k_obs_ t*])(1)
where ssDNA_tot_ is the overall concentration of DNA molecules with or without lesions. Typical unwinding curves with unmodified forked DNA substrates (5 nM) at four different RecQ concentrations (5–30 nM) are shown in Figure 4.

The pre-steady state, single turnover kinetics of unwinding of double-stranded DNA by RecQ, was extensively studied by Zhang et al. [9]. They showed that the unwinding step-size of RecQ was ~4 base pairs, while the unwinding rate determined in these pre-steady state, single turnover experiments; was ~84 bp/s. Under our experimental conditions, this fast-unwinding burst component is not time-resolved. The subsequent slow phase is dependent on the diffusion-controlled formation of RecQ DNA complexes that, once formed, undergo rapid reaction catalyzed by ATP. In the case of our DNA lesions, in contrast to modified DNA, the reactions of ATP with the RecQ DNA complexes is significantly greater than the observed unwinding rate (Appendix A). The effects of RecQ concentration on the slow-phase unwinding kinetics of unmodified DNA (5 nM) at four different RecQ concentrations (5–30 nM), are depicted in Figure 5 [9]. The values of *k_obs_* were determined from the best fits of Equation (1) to the second, slow phase (red lines) superimposed on the experimental data points.

The values of *k_obs_* increase linearly as a function of RecQ concentration (Figure 5), thus indicating that the unwinding rate is proportional to the rate of formation of helicase DNA complexes. The bimolecular association rate constant calculated from the slope of the linear plot in Figure 5 is *k_a_* = (8.8 ± 0.4) × 10^5^ M^−1^s^−1^. The rate constant *k_obs_* depends on the bimolecular encounter rate constant *k_a_* and the processivity *P* according to the equation [27]:*k_obs_* = *k_a_* [*C*_RecQ_]*P* where *P* = *k*_U_/(*k*_U_ + *k*_D_)(2)

The protein concentration is denoted by *C*_RecQ_, and the observed unwinding rate constant *k_obs_* is defined as (*k*_U_ + *k*_D_) [27]. The unwinding process is determined by the competition between unwinding and the dissociation rate of the RecQ DNA complex, defined by the rate constants *k_U_* and *k*_D_, respectively [9].

Adedeji et al. [28] considered the probability *p* that a helicase will successfully advance by one step to unwind the next set of base pairs, instead of dissociating from the DNA molecule. The probability of unwinding per bimolecular encounter is defined by the ratio *p* = *k*_U_/(*k*_U_ + *k*_D_) for each step. In the case of n-steps, the overall probability of fully unwinding a double-stranded DNA sequence is a product of individual probabilities:*P* = *p*_1_, *p*_2_, …, *p*_n,_ = [*k*_U_/(*k*_U_ + *k*_D_)]^n^(3)
with n representing the number of steps. This model provides the important insight that the value of *P* derived from plots of the fractions of unwound ssDNA products as a function of reaction time, can level off at a value of less than 1.0 if any single *p*_n_ value in Equation (3) is less than 1.0 for any of the steps in the double-stranded region. The presence of a DNA lesion in any of the n-steps will thus lower the overall processivity. The validity of Equation (3) was demonstrated experimentally using unmodified DNA substrates with double-stranded DNA sequences of different lengths [28]. In our experiments, the DNA lesions were embedded at the ninth nucleotide counted from the single/double strand junction of the forked DNA substrate (Figure 2). Thus, the RecQ helicase encountered the DNA lesions during the ~third step within the double-stranded region (Figure 2).

### 3.3. Unwinding DNA Containing Bulky DNA Lesions

Typical examples of the impact of the bulky DNA lesions on RecQ-catalyzed unwinding kinetics are shown in the following figures. These experiments were conducted at concentrations of 5 nM DNA substrate and 5 nM RecQ, which correspond to the concentrations of the unmodified DNA experiment shown in Figure 4A. The *k_obs_* values were calculated from the best fits of Equation (1) to the experimental data points. The reported *k_obs_* values represent averages of three independent measurements.

Except for the BP-A:T duplexes, none of the other BP-modified DNA samples exhibited burst signals at these low RecQ concentrations (5 nM).

#### 3.3.1. Intercalated (+)-trans-BP-A:T Adenine Duplexes

The BP-A adducts are derived from the binding of the BPDE residue to the exocyclic *N*^6^-amino group of adenine. The modified adenine residue is paired with T in the opposite DNA strand, and the BP aromatic ring system assumes an intercalative conformation without displacement of the modified adenine and is also flanked by normal undistorted base pairs on both sides [24]. The BP-modified adenine base BP-A exists in a major *syn-*glycosidic conformation instead of the normal B DNA *anti* conformation, which weakens the BP-A:T base pairing at the site of the adduct. However, a less abundant *anti* conformation was also detected, which suggests that the major *syn*-conformer exists in equilibrium with a minor *anti*-conformer [24]. This conformational flexibility is a property that might help the helicase to bypass this type of bulky BP-A DNA adduct.

The (+)-*trans*-BP-A:T adduct is the least DNA helix-distorting adduct as compared to all the guanine BP-G:C adducts studied in this work. Consistent with these physical properties, analysis of the (+)-*trans*-BP-A:T duplex unwinding curve (Figure 6) yields the *k_obs_* value (22.8 ± 3) × 10^−4^ s^−1^; this value represents a moderate ~50% reduction relative to the unmodified DNA *k_obs_* value of (49.0 ± 1) × 10^−4^ s^−1^ at the same RecQ and DNA concentrations (Figure 4A).

The *anti-syn* interconversion detected by NMR methods suggests that the flexible conformations characterizing the BP-A adduct may enhance the successful bypass of the (+)-*trans*-BP-A:T adduct by the RecQ helicase, thus enhancing the magnitude of *k_obs_*.

#### 3.3.2. Minor Groove (+) and (−)-trans-BP-G:C Adducts

The polycyclic aromatic ring systems of the (+)-*trans*- and (−)-*trans-*BP-G adducts are attached to the exocyclic *N*^2^-amino group of guanine (G) but with opposite orientations [20,21] relative to the 3′ → 5′ direction of the translocation of RecQ (Figure 3).

The kinetic unwinding curves for these two duplexes are shown in Figure 7. The *k_obs_* values are (1.22 ± 10^−4^) s^−1^ (−)-*trans*-, and (2.91 ± 0.1) × 10^−4^ s^−1^ (+)-*trans*-BP-G:C duplexes.

It was shown earlier that the (+)-*trans*-adduct causes a stronger and significantly more flexible DNA bend than the (−)-*trans* adduct which is more rigid [29,30]. The greater rigidity of the (−)-*trans-*BP-G-C and the greater conformational flexibility of the (+)-*trans-*adduct are correlated with the ~two-fold greater unwinding rate constant associated with the (+)-*trans*-BP-G:C adduct (Figure 7). The higher flexibility indicates that favorable helicase DNA conformations can be sampled more frequently, thus leading to a higher unwinding rate. Employing a different method that estimated the total yields of DNA unwinding products catalyzed by the 3′ → 5′ WRN helicase, Khan et al. observed similar (+)-*trans*/(−)-*trans* product ratios after fixed incubation time intervals [26].

#### 3.3.3. Base-Displaced Intercalation of the (+)-cis-BP-G:C Adduct

The (+)-*cis*-BP-G:C adduct is characterized by a base-displaced intercalative conformation (Figure 3) with the BP aromatic ring system inserted between adjacent base pairs, while the benzylic G deoxyguanosine ring is positioned in the minor groove with its plane parallel to the helix axis; the partner base C is looped out into the major groove [17]. The flanking base pairs are not disrupted and the BP aromatic ring system–base stacking interactions stabilize the intercalated BP-G:Del adduct conformations [16,31].

The unwinding curve is shown in Figure 8, and the *k_obs_* value of the (+)-*cis*-BP-G:C duplex is 1.42 *±* 0.22 s^−1^, which is close to the (−)-*trans*-BP-G:C value of *k_obs_* = 1.22 ± 0.1 s^−1^. These two adduct conformations are stereochemically related since both are characterized by absolute *R* stereochemistry about the BP-deoxyguanosine linkage site. This means that in both cases, the BP aromatic ring systems are positioned on the 3′-side of the modified guanine residues which sterically hinder the progress of the RecQ helicase translocating in the 3′→ 5′ direction. 

#### 3.3.4. Intercalated ‘Deletion’ Duplexes, BP-G:Del

Deletion duplexes (Del) are identical to the full duplexes discussed up till now, except for the deleted canonical Watson–Crick C nucleotide opposite the BP-G modified guanine residue (abbreviated as G:Del). The Del duplexes containing DNA lesions occur *in vivo* and, if not removed by cellular DNA repair systems, can contribute to mutagenesis [32,33]. In contrast to the minor groove conformations of the (+)- and (−)-*trans*-G:C duplexes, in G:Del duplexes the same adducts are fully intercalated between adjacent base pairs without displacement of the modified guanine residues from their usual positions [18,34].

#### 3.3.5. Intercalated (+)- and (−)-*trans-*G:Del Duplexes

The *k_obs_* value of the (+)-*trans*-BP-G:Del adduct (4.33 ± 0.21) × 10^−4^ s^−1^ (Figure 6) is modestly enhanced by a factor of 1.5 relative to the full (+)-*trans*-BP-G:C duplex *k*_obs_ value. However, in the case of the (−)-*trans*-BP-G:Del adduct, the (−)-*trans-*BP-G-Del *k*_obs_ value is (8.35 ± 0.6) × 10^−4^ s^−1^, which is ~seven times greater than the (+)-*trans*-BP- G:C *k*_obs_ value.

This change is consistent with the minimal structural distortions of the G:Del duplex that is characterized by, a thermodynamically stabilized intercalative conformation [23], in contrast to both *trans*-BP-G:C duplexes that are thermodynamically less stable [16]. It is remarkable that the deletion of a single nucleotide in the unmodified partner strand causes a five-fold increase in the unwinding rate constant *k_obs_.*

#### 3.3.6. Intercalated (+)-*cis*-BP-G:Del Deletion Duplexes

In the case of the (+)-*cis*-BP-G:Del adduct, the BP aromatic ring system remains intercalated as in the (+)-*cis*-BP-G:C duplex [35]. The structural features of the (+)-*cis*-BP-G:Del duplex resemble those of the full (+)-*cis*-BP-G:C duplex [35]. The BP aromatic ring system adopts a similar wedge-shaped intercalated structure, but the modified deoxyguanosine base is displaced into the minor groove with its plane aligned parallel to the DNA helical axis.

Particularly noteworthy is the large increase in the *k_obs_* values from 1.42 *±* 0.22 s^−1^ in the full (+)-*cis*-BP-G:C duplex, to 5.99 × 10^−4^ s^−1^ in the (+)-*cis*-BP-G:Del duplex (Figure 8).

The structural features of the (+)-*cis*-BP-G:Del duplex are almost identical to those of the full (+)-*cis*-BP-G:C duplex, except for the absence of the cytosine in the (+)-*cis*-BP-G:Del duplex. Indeed, the only apparent difference between the full (+)-*cis*-BP-G:C duplex and the (+)-*cis*-BP-G:Del duplex is the presence of the displaced cytosine positioned in the major groove of the full (+)-*cis*-BP-G:C duplex [17]. This orphaned and unpaired cytosine residue appears to play a major role in diminishing the unwinding rate of the full duplex relative to the (+)-*cis*-BP-G:Del duplex. 

In summary, the *k_obs_* values of all DNA adducts discussed up till this point are compared to one another in Figure 9.

## 4. Non-Bulky DNA Lesions

### 4.1. Unwinding DNA Containing an Oxidative DNA Lesion

Spiroiminodihydantoin (Sp, *S* stereoisomer) is an oxidation product of 8-oxoguanine in DNA [36] that is generated in vivo under conditions of oxidative stress associated with the inflammatory response [37]. Thermodynamic, NMR, and molecular dynamics simulation studies indicate that the propeller-like twisted Sp ring structure resides in the major groove of double-stranded DNA and causes a significant destabilization of the DNA duplex [38]. The *k_obs_* value determined from the Sp unwinding curve (Figure 10) is (13.4 ± 0.9) × 10^−4^ s^−1^.

### 4.2. Cis-Syn Cyclobutanepyrimidine (Thymine) Dimer (CPD) and the T(6−4)T UV Irradiation Products

In the case of CPD, all Watson–Crick base pairs remain intact, but the helix is bent by 9^0^ at the site of the lesion. The T(6−4)T lesion is significantly distorted with the two cross-linked thymine bases oriented with their planes perpendicular to one another; the helix is bent by 44°, and hydrogen bonding is absent at the 3′-flanking base pair [39,40]. The *k_obs_* values of the CPD and T(6−4)T lesions are (24.6 *±* 1) × 10^−4^ and (21.2 ± 2) × 10^−4^, respectively (Figure 10); within experimental error, these values are the same as the bulky (+)-*trans*-BP-A:T value, while the Sp value is ~40% smaller (Figure 9). These observations alone underscore the conclusion that the bulk of the DNA lesion alone does not seem to affect the magnitudes of *k_obs_* unwinding rate constants. However, as discussed below, the altered chemical structures of the nucleobases in these non-bulky DNA lesions are the dominant factors that affect the processivities *P.*

Only one nucleotide is modified in the case of Sp, as well as in all of the bulky DNA adducts, and the (+)-*trans-*BP-A:T adenine duplex. The DNA unwinding characteristics of all bulky DNA adducts studied indicate that they diminish the rates of unwinding, but not the processivities, since *P* = 1.0 in all cases. While the unwinding rates of duplexes are slowed to different extents by different BP-G:C adducts, all are eventually fully unwound. In the case of the non-bulky DNA lesions studied, *P* < 1.0, and full unwinding is not achieved. The Sp processivity *P* = 0.63 is significantly smaller than the *P* = 1.0 value of the bulky (+)-*trans-*BP-A:T and BP-G:C adduct duplexes. In the case of CPD, *P* = 0.11, and only 0.062 in the case of the T(6−4)T lesion (Figure 10).

These observations suggest that the altered chemical structures of these three non-bulky lesions and the associated structural distortions of one (Sp), or two nucleobases (CPD and T(6−4)T), dominate the mechanism of inhibition.

The processivity is defined as the probability that a helicase will proceed to the next unwinding step rather than dissociating from the DNA. Since the processivity *P* = *k_U_/k_ob_*_s_, and *k_ob_*_s_ = *k_U_* + *k_D_*, a decrease in *k_U_* must be accompanied by a proportional increase in *k_D_* since the *k_obs_* values remain constant. The strong reductions of the *P* values, but not the unwinding rate constants (Figure 9), are due not only to the reduction in unwinding rate constant *k_U_*, but also a strong enhancement of the dissociation constant *k_D_* relative to *k_U_* (Equation (3)). The unwinding rate constants (*k*_obs_) are not significantly smaller than the rate constants associated with the bulky DNA lesion, including the (+)-*trans*-A:T duplex. It is noteworthy that *P* decreases in value in the case of the non-bulky DNA lesions, while the *k_obs_* values do not change significantly. These results can be explained by considering that a decrease in *k_U_* means that the residence time of the helicase bound to its DNA substrate becomes longer. In turn, the longer residence time suggests that the probably of dissociation of the helicase per step must also increase, thus leading to an increase in *k_D_*, and a lower processivity *P*. Thus, there is a correlation between decreasing *k_U_* and increasing *k_D_* values in this case.

The reason for the lower processivity and diminished *k_U_* rate constants of the two UV photolesions may be due to the fact the helicase needs to bypass two nucleotides rather than only one in the case of Sp and the bulky DNA adducts. In the case of the bulky polycyclic aromatic adducts, the same phenomenon does not manifest itself, possibly because of non-covalent Van der Waals interactions between the helicase and bulky polycyclic aromatic ring systems that diminish the helicase dissociation rate constant *k_D_*. In the case of Sp, such interactions are most likely minimal because of its non-aromatic nature and small bulk (Figure 1B).

## 5. Summary and Conclusions

The unwinding rates of double-stranded DNA catalyzed by the 3′ → 5′ translocating *E. coli* helicase RecQ is inhibited by bulky, benzo[*a*]pyrene diol epoxide-derived DNA guanine and adenine adducts (BP-G and BP-A). In the case of the adenine adduct, intercalation of the bulky BP polycyclic aromatic ring system does not strongly disrupt either base pairing or the normal B-DNA structure; the unwinding rate constant is diminished by a factor of ~two only, relative to unmodified DNA.

In the case of the bulky guanine adducts (BP-G:C), the BP aromatic ring system adopts two types of adduct conformations: minor groove, or intercalation with displacement of the modified guanine and partner C residues into the major or minor grooves of B-DNA. The strong distortions of the normal B-DNA conformations in both cases result in up to ~50-fold decreases in unwinding rate constants. However, all BP-G:C duplexes, regardless of BP-G:C adduct conformation, can be fully unwound (processivity *P* = 1.0).

The lowest unwinding processivities are observed in the case of the non-bulky guanine oxidation product Sp, and the UV-radiation-induced products CPD T^^^T and T(6−4)T (Figure 1). However, their unwinding rate constants (*k*_obs_) are mostly and significantly greater than the rate constants associated with most of the bulky DNA adducts (Figure 9). Helicase unwinding is a multi-step process, and the RecQ helicase step size is four base pairs per step. In the case of the CPD and T(6−4)T dimeric lesions, the helicase must bypass two structurally distorted nucleobases out of four, rather than just one, thus accounting for the lowest processivities that characterize these two lesions. These observations can be accounted for by correlated decreases in unwinding rate constants and enhancements in the helicase DNA complex dissociation rate constants.

## Figures and Tables

**Figure 1 ijms-23-15654-f001:**
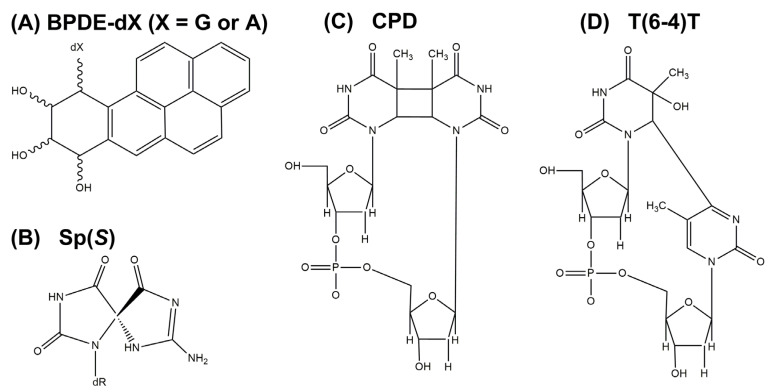
Structures of DNA lesions. (**A**) The BPDE-*N*^2^-dG or -*N*^6-^dA adduct. (**B**) SP(*S*), Spiroiminodihydantoin (*S* stereochemistry). (**C**) Cyclobutanepyrimidine thymine dimer. (**D**) pyrimidine(6−4)pyrimidone thymine dimer.

**Figure 2 ijms-23-15654-f002:**
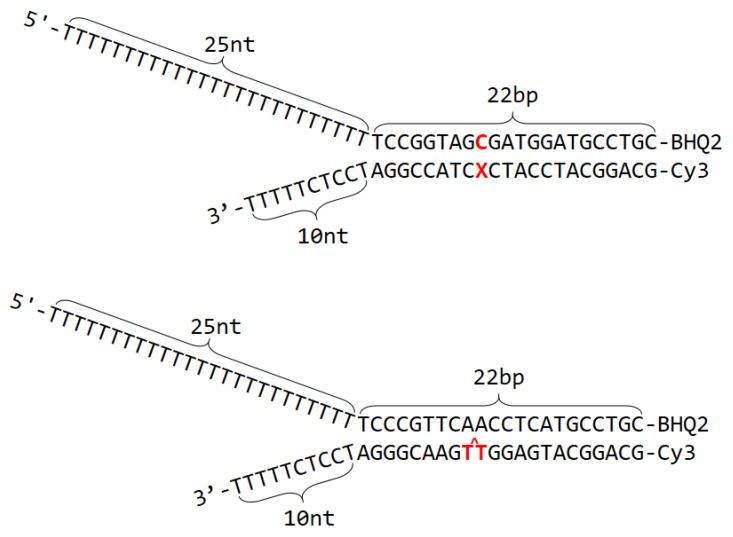
Definition of the forked, unmodified DNA helicase substrate.

**Figure 3 ijms-23-15654-f003:**
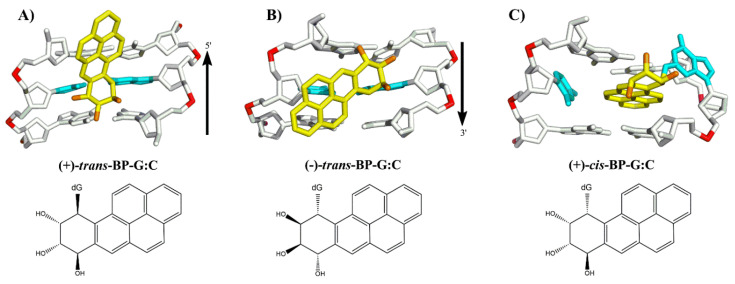
Conformations of stereoisomeric benzo[*a*]pyrene diol epoxide—derived *N*^2^-guanine adducts (G) in double-stranded DNA. (**A**) (+)-*trans*-BP-G:C; (**B**) (−)-*trans*-BP-G:C; (**C**) (+)-*cis*-BP-G:C. All three BP-modified guanine residues are paired with cytosine (**C**) in the opposite strand.

**Figure 4 ijms-23-15654-f004:**
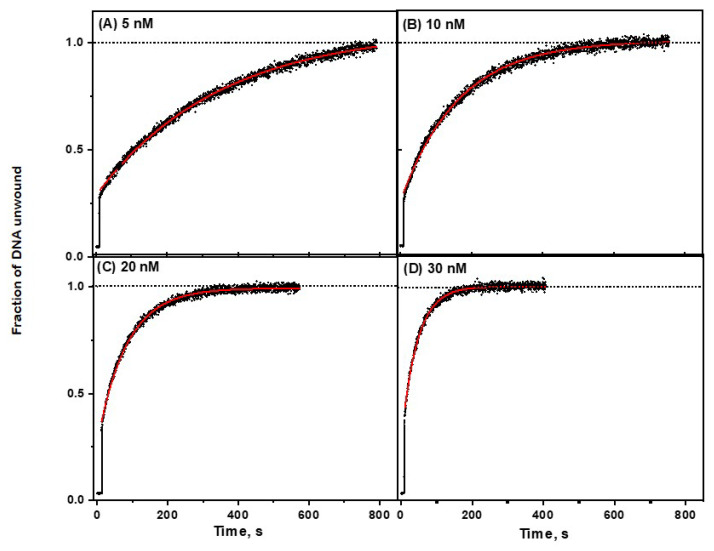
Effect of RecQ concentration on the unwinding kinetics of unmodified DNA substrates (Figure 1). [DNA] = 5 nM. Fits of Equation (1) (red lines) to the experimental data points. The following optimized *k_obs_* values were obtained at different RecQ concentrations with standard errors representing 95% confidence intervals: (**A**) 0.0049 ± 0.0001 s ^−1^ (5 nM), (**B**) 0.0087 ± 0.0001 s ^−1^ (10 nM), (**C**) 0.0193 ± 0.0001 s ^−1^ (20 nM), and (**D**) 0.0261 ± 0.0002 s ^−1^ (30 nM).

**Figure 5 ijms-23-15654-f005:**
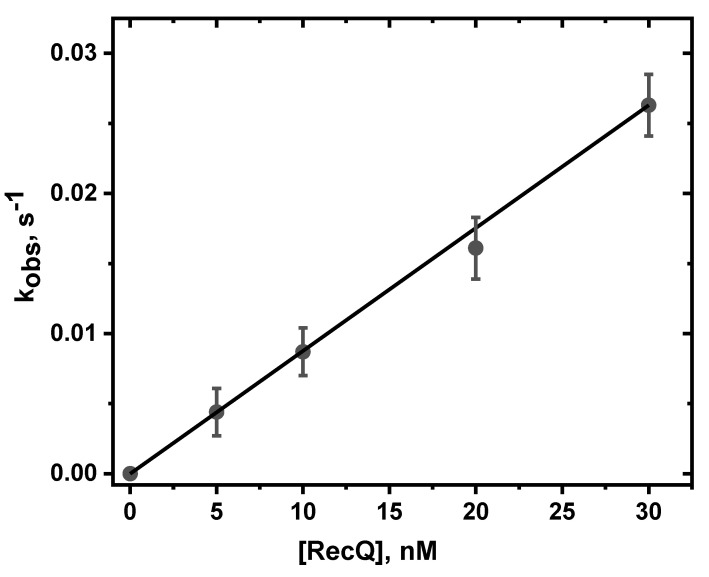
Effect of RecQ concentration on the observed rate constant *k_obs_*. The slope of the straight line was determined by a least squares fit to the data points with a standard error representing a 95% confidence interval. This slope is equal to the rate constant *ka* (um DNA) = (8.8 ± 0.4) × 10^5^ M^−1^ s^–1^.

**Figure 6 ijms-23-15654-f006:**
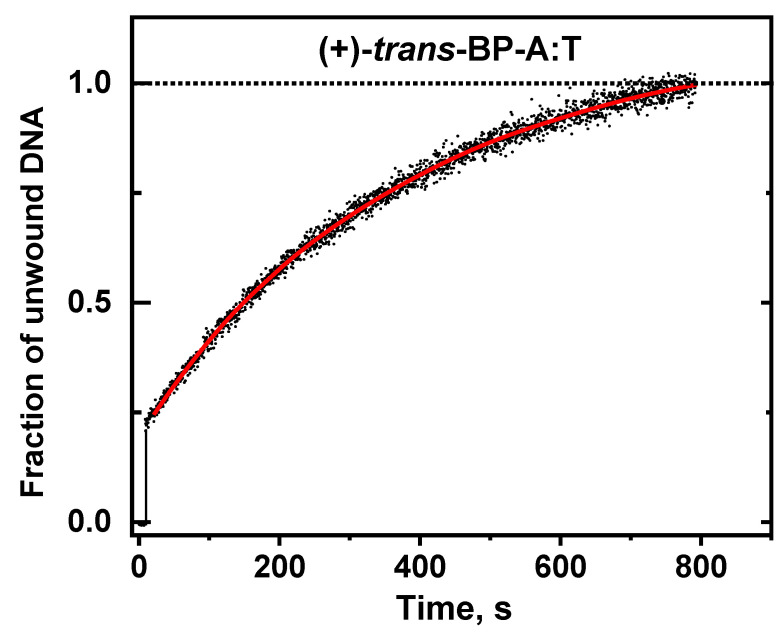
DNA unwinding kinetics of DNA substrates with single bulky (+)-*trans*-BP-A:T adduct base pairs.

**Figure 7 ijms-23-15654-f007:**
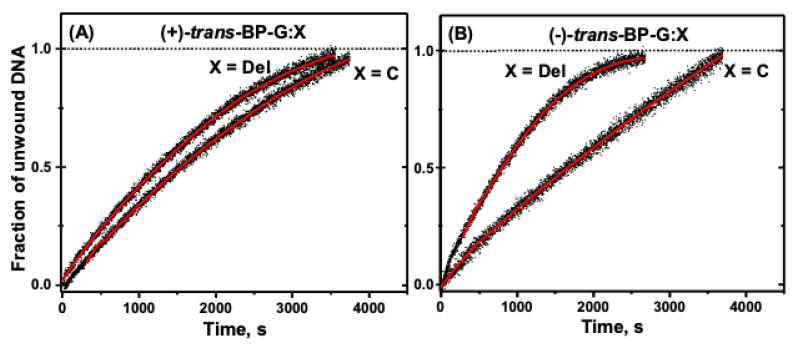
(**A**) Unwinding curves of (+)-*trans*-BP-G:C and (+)-*trans*-BP-G:Del duplexes. (**B**) Unwinding curves of (−)-*trans*-BP-G:C and (−)-*trans*-BP-G:Del duplexes.

**Figure 8 ijms-23-15654-f008:**
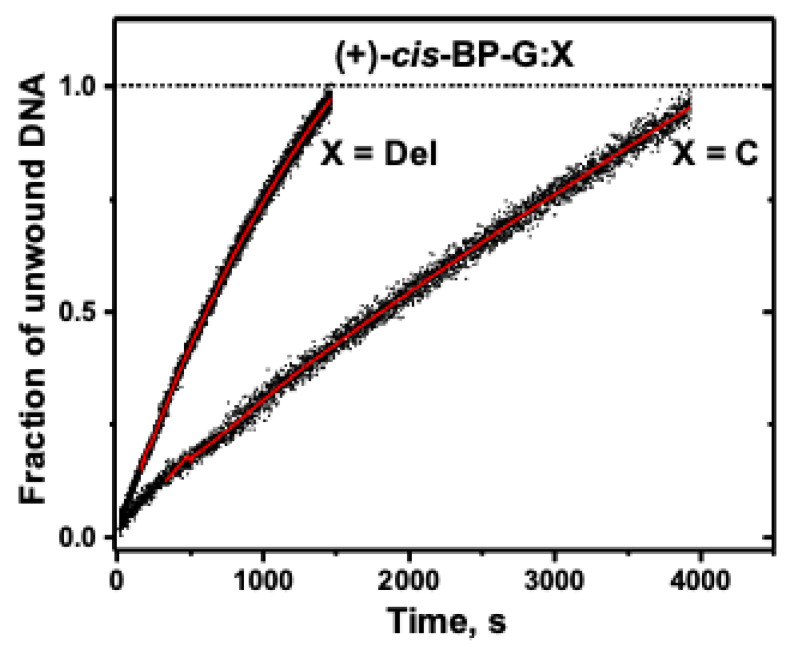
Unwinding curves of (+)-*cis*-BP-G:C and (+)-*cis*-BP-G:Del duplexes.

**Figure 9 ijms-23-15654-f009:**
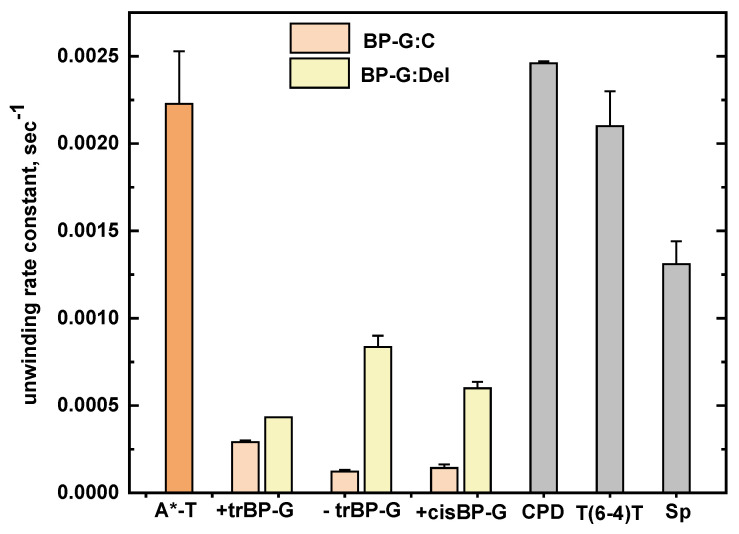
Comparisons of unwinding rate constants (*k_obs_*). The unmodified DNA value (0.00049 ± 0.0001 s^−1^) is not shown in this graph.

**Figure 10 ijms-23-15654-f010:**
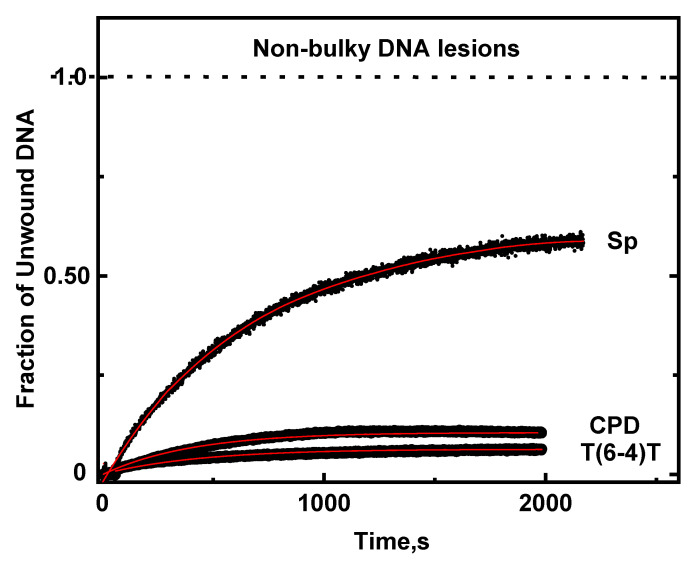
Unwinding curves of DNA duplexes containing single Spiroiminodihydantoin (Sp), the CPD thymine dimer, and the T(6−4)T UV thymine dimer photoproducts. Burst signals are not shown but comprise the following fractions of DNA duplexes: CPD: 19%, T(6−4)T, 30%; Sp, no burst.

## Data Availability

Not applicable.

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
