# Peer review of "Inhibition of *E. coli* RecQ Helicase Activity by Structurally Distinct DNA Lesions: Structure—Function Relationships"

_ijms, 2022, doi:10.3390/ijms232415654_

Round 1

Reviewer 1 Report

The article from Sales et al. presents a careful look into the correlation between different types of DNA lesions and the function of E. coli RecQ helicase in unwinding dsDNA containing these lesions. The authors used fluorescence-based kinetics methods to characterize the unwinding of dsDNA by measuring recovery in quenched fluorescence. While the study is important in understanding RecQ related DNA unwinding with different lesions in DNA site, in its current form, the manuscript presents many serious flaws and cannot be recommended for acceptance in its current form in the ijms journal.

Here are some comments that the authors can work on to improve the quality of the manuscript:

1.      The work lacks careful experiment design and hypothesis that the authors want to test. Going over the manuscript, it looked like the author wanted to test the role of different lesion structures (which is different from their conformation as cis/trans). However, the experiment design mix both conformations and structures and do not quite convey how individually they impact the RecQ activity. Mixing them and comparing them together (as in figure 7) creates lot of confusion into how these lesions are correlated with RecQ unwinding activity.

2.      The authors essentially missed the first unwinding component occurring during the burst phase as “non-time resolved”, skewing the kinetics towards a slow diffusion-controlled component. Are there other ways to characterize this fast component (its percentage as compared to slow component) as it clearly presents within unmodified as well as some modified DNA?

3.      Why the fitting is shown for fraction of DNA unbound but not for modified DNA? Do all modified DNA follow a single exponential kinetics?

4.      The figures' titles are not quite sufficiently explained (for e.g figure 1 and 2). The sequence of figures and their introduction in the text is not aligned properly and sometimes perhaps misquoted. For e.g in page 3, line 86, it is perhaps figure 2 instead of figure 1. Also in page 6, fig 3 description, the authors are perhaps recalling figure 2 instead of 1.

5.      Page 9, line 245-246; In contrary to the statement, the kobs values for (+)-cis-BP-G*C and (-)-trans-BP G*del looks very different and not close at all.

6.      The order of sections and figures are extremely confusing. Section 3.1 refers to figure 4 and 3.2 refers to figure 3.

7.      Page 3, line 89-90. Not quite clear what is glycerol MgCl2.

8.      The manuscript lacks details on the raw data, fitting and analysis done by the authors, which could be presented in supporting information to improve the quality of work for the readers.

9.      In figure 7, the authors provide no details on how the error analysis is done. Are the errors reported calculated from different experiments? If so, how many repeat experiments are done?

10.   No error bar reported for +transG*del

Author Response

Inhibition of E. coli RecQ Helicase Activity by Structurally Distinct DNA Lesions: Structure – Function Relationships.  A.H. Sales et al.

Reviewer 1 Corrections.

We thank the reviewer for the astute analysis of our previously submitted article. As a result of these valuable comments, we concluded that we needed to reorganize and rewrite the article  for better clarity.  We therefore reorganized  and rewrote the Results and Discussion section and considered the reviewer’s specific comments one by one.

  1. (a) ….Lacks experimental design and hypothesis, difference between structure and conformation.

Structures: We used a family of enantiomeric (+ or -) benzo[a]pyrene-derived (BP) diol epoxide -DNA adducts all have similar but stereochemically different  basic structures as summarized below:

              (b)  Adduct conformations of stereoisomeric adducts are shown in Figure 3.

  1. Additions/corrections: Introduction, 2nd Paragraph (lines 51-68).Overview: relevant to design, lines 130-143. Objectives and design; 417-438.  Objectives and summary of results. Lines 417-438.

  1. Further clarifications implemented:

(i) The three stereoisomeric bulky aromatic ring system-N2-guanine adducts adopt different adduct conformations in DNA .

(ii) the stereochemical substituents of the non-aromatic benzylic rings are oriented  differently relative to one another (defined as structural stereoisomers). 

(iii) Conformations: Each stereoisomeric adducts adopts a different conformation in DNA (Figure 3).

 (stereochemistry of each is identified.

(iii) Hypothesis: the different conformations (such as intercalation with and without base displacement, or external  minor groove conformations) would affect the helicase unwinding activities (unwinding rates and/or processivities)

Text changes:  Introduction, Lines 44-50, while the goals are summarized in the Overview  section, lines 130-143.

  1. Burst phases not shown.

Unmodified DNA and one bulky adduct (the adenine adduct BP-A*:T exhibit burst phases. The non-bulky DNA adducts that did exhibit bursts, are now specified in the legend to Figure 10 (these bursts could not be added to Figure 10 because the vertical lines would overlap  one another; instead, the magnitudes of the burst are stated in the Figure 10 legend.

  1. Curve fits for DNA with lesions/adducts not shown. All the single-exponential fits are now shown  (red linessuperimposed on the experimental data).  

  1. Mislabeled references to Figures. These typos have now been fixed.

  1. (-)-trans and (+)-cis-BP-G*:C duplexes, similar unwinding rate constants…….. Yes you are right. Thank you for your careful reading. This point is now correctly explained in section 5.3, lines 305-310.

  1. Order of sections confusing……..this problem has, hopefully, been fixed because each adduct or lesion, is now discussed in separate sections that are clearly labeled and contain the correct figure references.  

  1. MgCl2…now fixed. Line 111.

  1. …….the manuscript lacks details on the raw data. We now present all the raw data (kinetics of unwinding and curve fitting (the fits of Eq. 1 are shown as red lines).

  1. Reproducibility. The rate constants reported are averages of three independent experiments (added to the text)

  1. No error bar reported for (+)-trans G*:Del. We now explicitly report all the numerical kobs unwinding rate constants in the text.

  1. Small (+)-trans-G*:Del error bar…… we now report the values and error bars of the constants (see line 322, section 5.4.1.).   The error bar is ~ 5% and does not show well in the bar graph because the kobs value is small too!

Thank you for your careful reading and comments that helped us to produce a much-improved manuscript.

A.H. Sales et al.

Reviewer 2 Report

The manuscript is potentially interesting however full of typos and spelling errors. Therefore, authors must consider correcting the paper in its entirety in order to be reconsidered for publication.

Here are some examples:

1)    Line 1. “China” has a different font / size than the rest of the text, as well as the abstract and introduction seem to have a different size.

2)    Line 21. After the parenthesis there is a comma, presumably in place of the period.

3)    Line 16, and along the whole manuscript, there is an extra space.

4)    Line 26, 31, 52 and along the whole manuscript, the period is before the parenthesis instead of after.

5)    Line 47, there is an extra period in the center of a sentence without any valid reason.

6)    Line 51, 52, 22 and along the whole manuscript spaces missing.

7)    Line 90. MgClis reported twice in the description of the composition of the buffer and moreover one of the two times without the concentration values.

Unfortunately, these are just some examples of the errors reported in the paper, therefore to be revised it must necessarily be corrected again by the authors.

Author Response

Inhibition of E. coli RecQ Helicase Activity by Structurally Distinct DNA Lesions: Structure – Function Relationships.  A.H. Sales et al.

Reviewer 2

We thank the reviewer for pointing out these important formatting problems.  We have carefully gone over our formatting errors and have made the corrections needed.

1, Specifically, the format of the citations in the text are now properly spaced.

  1. The correct spaces between text and citations have been introduced.
  2. Periods are placed after the parentheses.
  3. Extra spaces have been removed.
  4. Missing spaces have been introduced.

Thank you for helping us to correct these errors.

A.H. Sales et al.

Round 2

Reviewer 1 Report

The authors have taken my comments into consideration and reorganized the manuscript to improve readability and scientific soundness. I have no further questions for the authors and recommend this work for publication.

Reviewer 2 Report

The manuscript can be accepted in the present form.